# Real-Time Measurement of Drilling Fluid Rheological Properties: A Review

**DOI:** 10.3390/s21113592

**Published:** 2021-05-21

**Authors:** Naipeng Liu, Di Zhang, Hui Gao, Yule Hu, Longchen Duan

**Affiliations:** 1School of Automation, China University of Geosciences, Wuhan 430074, China; lnp@cug.edu.cn; 2Hubei Key Laboratory of Advanced Control and Intelligent Automation for Complex Systems, Wuhan 430074, China; 3Engineering Research Center of Intelligent Technology for Geo-Exploration, Ministry of Education, Wuhan 430074, China; 4School of Earth Resources, China University of Geosciences, Wuhan 430074, China; zd@cug.edu.cn; 5Faculty of Engineering, China University of Geosciences, Wuhan 430074, China; gaohui@cug.edu.cn (H.G.); ylhu@cug.edu.cn (Y.H.)

**Keywords:** drilling fluid, rheological properties, real-time measurement, pipe viscometer, Couette viscometer, artificial intelligence

## Abstract

The accurate and frequent measurement of the drilling fluid’s rheological properties is essential for proper hydraulic management. It is also important for intelligent drilling, providing drilling fluid data to establish the optimization model of the rate of penetration. Appropriate drilling fluid properties can improve drilling efficiency and prevent accidents. However, the drilling fluid properties are mainly measured in the laboratory. This hinders the real-time optimization of drilling fluid performance and the decision-making process. If the drilling fluid’s properties cannot be detected and the decision-making process does not respond in time, the rate of penetration will slow, potentially causing accidents and serious economic losses. Therefore, it is important to measure the drilling fluid’s properties for drilling engineering in real time. This paper summarizes the real-time measurement methods for rheological properties. The main methods include the following four types: an online rotational Couette viscometer, pipe viscometer, mathematical and physical model or artificial intelligence model based on a Marsh funnel, and acoustic technology. This paper elaborates on the principle, advantages, limitations, and usage of each method. It prospects the real-time measurement of drilling fluid rheological properties and promotes the development of the real-time measurement of drilling rheological properties.

## 1. Introduction

In the drilling industry, almost every step requires drilling fluid [1]. The drilling fluid’s properties have a significant impact on drilling efficiency and safety. The successful completion and cost of an oil well depend mainly on the drilling fluid’s performance. The cost of the drilling fluid itself is relatively small, but the choice of the right drilling fluid program and maintenance of fluid properties while drilling profoundly influence the total well costs [2]. The cost of drilling fluid accounts for 5% to 15% of the entire drilling cost, but it can solve 100% of drilling problems [3]. The physical and chemical properties of drilling fluid, such as its density and rheological properties, have a significant impact on the processing and control of well conditions [4,5]. A high-viscosity drilling fluid is desirable to transport cuttings from downhole up to the surface and suspend weighting agents (such as barite). However, if the viscosity is too high, the friction is high, which may hinder the circulation of the mud, resulting in excessive pump pressure, reducing the drilling speed and hindering the solids removal equipment [3].

The drilling fluid properties play an important role in the optimization of the rate of penetration. In the rate of penetration models established by many scholars, such as mathematical and physical models [6,7] and artificial intelligence models [8,9,10,11,12,13], the drilling fluid’s properties are the influencing factors. Therefore, real-time optimization of the drilling fluid’s performance can increase the rate of penetration, while measuring the drilling fluid’s properties in real-time is a prerequisite.

Although the automation of drilling has progressed over the past decade, the current technological control of drilling fluid quality was established 50 years ago and is essentially manual. The American Petroleum Industry’s (API) standard test manual for the petroleum industry requires a complete drilling fluid measurement program to be performed twice a day. At the drilling site, the Marsh funnel is usually used to quickly test the viscosity of the drilling fluid, but this Marsh funnel viscosity can only roughly estimate the viscosity of the drilling fluid, and it cannot reflect all of the drilling fluid’s rheological properties. The rotational Couette standard viscometer can measure all of the drilling fluid’s rheological properties, but they need to be measured in a laboratory after sampling at the site is conducted. If the drilling fluid’s performance does not meet the requirements, it is necessary to test to determine the additive amount to ensure the maintenance of the drilling fluid’s performance. This sample testing and data analysis may take several hours [14]. During the drilling process, the drilling fluid’s properties will change due to the addition of substances in the formation. The optimum management of drilling fluid maintenance requires frequent, accurate, and reliable measurements of mud properties. If the fluid properties of the drilling cannot be acquired when the formation changes, the drilling security is greatly threatened. Therefore, real-time measurement can diagnose and adjust the drilling fluid’s performance immediately [15,16,17]. The real-time measurement of drilling fluid can also promote the automation process of drilling fluid control [18]. In summary, the current drilling fluid measurement technology cannot meet the needs of real-time measurement. It is necessary and urgent to achieve real-time measurement of drilling fluid properties.

## 2. Rheological Properties of Drilling Fluid

The drilling fluid’s rheological properties refer to the characteristics of flow and deformation under the action of external force. The drilling fluid’s rheological properties are essential for the following determinations: estimation of hole cleaning efficiency, calculation of frictional pressure losses in pipes and annuli, determination of equivalent circulating density (ECD) under downhole conditions, determination of prevailing flow regime in pipes and annuli, estimation of swab and surge pressures, and hydraulic optimization for improved drilling efficiency. Proper rheological properties can carry bottom hole cuttings to the ground quickly, increase the rate of penetration, reduce power consumption, ensure drilling safety, and improve economic benefits. In order to estimate the above-mentioned functions of the drilling fluid in time, it is important to measure the drilling fluid rheological properties in real time.

Figure 1 depicts how fluids can be classified based on their rheological behavior. There are Newtonian and non-Newtonian fluids. The viscosity of a Newtonian fluid does not change under different shear rates, while the viscosity of a non-Newtonian fluid changes under different shear rates (e.g., water, salt solutions, and light oil) [2]. The behavior of non-Newtonian fluid mainly includes three types: dilatant behavior (e.g., water-based polyethylene glycol drilling fluid [19]), pseudoplastic behavior (e.g., drilling fluid with high clay content, high waxy crude oil, and paint), and plastic behavior (e.g., aqueous solutions and emulsions of polymer compounds). The viscosity of the dilatant fluid increases with the increase in the shear rate, while the viscosity of the pseudoplastic fluid and the plastic fluid decreases with the increase in the shear rate. Plastic fluid starts to flow after a given shear stress is applied, while pseudoplastic fluid can flow under any shear force. Drilling fluids rarely exhibit dilatant behavior, as most of the drilling fluids used are plastic and pseudoplastic fluids. Through the rheological curve of the drilling fluid, we can more accurately evaluate the ECD and the ability of carry cuttings, optimize the hydraulic parameters, etc.

The most common models used to describe the drilling fluid rheology in the petroleum industry are the Bingham plastic model (Equation (1)), Power law (PL) model (Equation (2)), and Herschel–Bulkley model (Equation (3)):(1)τ=YP+PV∗γ
(2)$$\tau =K\gamma^{n}$$
(3)$$\tau =\tau_{0}+K\gamma^{n}$$
where YP is yield point, PV is the plastic viscosity, *τ*_0_ is the fluid yield stress, *K* is the consistency factor, *n* is the flow behavior index, and *γ* is the shear rate. The total ability for a Bingham plastic fluid to resist flow could be expressed by an apparent viscosity (AV) or effective viscosity for a given shear stress [2]. Generally, the shear rate of apparent viscosity is 1021 s^−1^ in API.
(4)$$\mathrm{AV}=\frac{\tau }{\gamma }=\mathrm{PV}+\frac{\mathrm{YP}}{\gamma }$$

The Herschel–Bulkley model is used when the accuracy of the rheological parameter measurement is high or in laboratory research. The API recommends using this model; it consistently provides good simulations of measured rheological data for both water-based and non-aqueous drilling fluids. For this reason, it has become the de facto rheological model for engineering calculations in the petroleum industry [2].

## 3. Real-Time Measurement Technologies

The current measurement of rheological properties utilizes a typical manually controlled rotational Couette viscometer. In recent years, many scholars have researched real-time measurement of drilling fluid rheological properties. The real-time measurement technology of drilling fluid rheology mainly includes the following four methods: (1) online rotational Couette viscometer, (2) pipe viscometer, (3) mathematical and physical model or artificial intelligence (AI) model based on a Marsh funnel, and (4) tuning fork technology.

### 3.1. Online Rotational Couette Viscometer

The recommended methods for drilling fluid analysis are presented in API 13B [20]. The drilling fluid’s rheological properties are measured using a standard rotational Couette viscometer as shown in Figure 2. The drilling fluid is placed in the measurement chamber, the motor drives the rotor sleeve to rotate at a constant speed through the transmission device. The viscosity of the measured liquid acts on the bob to generate a certain torque that drives the torsion spring, which is connected to the bob to produce an angle (dial reading). The dial reading is proportional to the viscosity of the fluid; thus, the viscosity of the fluid is calculated by the measured value of the dial reading. The standard rotational Couette viscometer has six rotation speeds: 3, 6, 100, 200, 300, and 600 r/min, related to shear rate: 5.11, 10.21, 170.2, 340.3, 510.5, and 1021 s^−1^.

The parameters of the Bingham model (PV, YP, and AV) [19], PL model (*n* and *K*), Herschel–Bulkley model (*τ*_0_, *n*, and *K*), and 10 s and 10 min gel strength can be obtained using a standard rotational Couette viscometer. Many scholars developed an online rotational Couette viscometer based on a standard rotational Couette viscometer to measure the drilling fluid’s rheological properties. The main improvement method is to add a control circuit to control the speed of the motor, turn the reading into an electrical signal output, and add the drilling fluid pipeline to automatically fill the measurement chamber with the drilling fluid. Figure 3 shows a typical online rotational Couette viscometer schematic diagram.

Saasen et al. [21,22] developed an online rotational Couette viscometer. The control box is used to change the rotational speed; shear stress readings were collected and transmuted to the acquisition device. In order to calibrate the drilling fluid’s temperature response over a suitable temperature range, a temperature sensor was attached to the measuring unit. The measurements of the online rotational Couette viscometer were compared to and corresponded well with the standard rotational Couette viscometer. Broussard et al. [23] also developed an online rotational Couette viscometer and conducted experiments in water-based and oil-based mud wells. It demonstrated the possibilities of automated drilling fluid measurements by using an online rotational Couette viscometer. However, there were some shortcomings such as that the viscometer was easily plugged up by gels particles and solids. No data were recorded during the brief testing of the water-based fluid at the first well due to the revealed heavy buildup of gel particles within the unit. During the first oil-based mud well, the automated sensor, equivalent to 600 rpm, measurements were consistently higher than the standard rotational Couette viscometer measurements. The 6 and 3 rpm equivalent measurements taken by the online viscometer trended very well when compared to the standard viscometer. For the second well, the online viscometer measurements for the water-based mud were significantly lower than the corresponding standard viscometer in the measurements below 300 rpm. Rheological property measurements taken by the online viscometer had an increased amount of noise in the measurements due to the significant accumulation of solids on the equipment. In order to avoid solids accumulation, Stock et al. [24] and Ronaes et al. [25] used a built-in pump and valve system for fluid sampling and cleaning cycles.

Magalhães et al. [26,27] used a rheology measurement instrument obtained by a modifying a Brookfield process viscometer, TT-100. The operational condition limits of the device were 1 to 15 bar (14.7 to 220.5 psi) of total pressure, temperature up to 160 °C (256 °F), and volumetric flow rate between 1 and 3 m^3^/h. There were four types of drilling fluid tested, namely, Newtonian fluid, pseudoplastic fluid, water-based drilling fluid, and synthetic-based drilling fluid. The measurement results showed that Newtonian fluid, represented by glycerin, and pseudoplastic fluid, represented by CMC solution, were statically consistent in terms of curve fitting. The results of the water-based mud and non-aqueous drilling fluid found some divergences. Compared with the standard viscometer measurement, the TT-100 tended to underestimate the shear stress in water-based mud, while in the non-aqueous drilling fluid, TT-100 overestimated the shear stress. The limitation of the viscometer was the size of the solids suspended. This article points out that the maximum diameter of the solids in the drilling fluid tested by the instrument must be less than 1 mm. Magalhães et al. [28] installed it on an onshore rig site near the northeast of Brazil, and operated it for several weeks. The proposed device and methodology for measuring online rheology produced similar results to the standard offline technique.

Dotson et al. [29] used a different rotor-bob geometry Couette viscometer from the standard rational Couette viscometer. The relevant non-Newtonian correction factor was applied to agree with the standard rational Couette viscometer reading. Rheological property measurements were performed by batch analysis of the fluid every 10 to 60 min. Once the previous sample was fully displaced, the inlet and outlet valves on the respective flowlines were closed. The sample was then pressurized to 0.55 to 0.69 MPa to collapse large air/bubbles in the fluid and to help ensure the viscometer was filled. Then, the fluid was agitated and heated to a user-defined temperature, typically 120 °F (48.9 °C) or 150 °F (65.5 °C). After each rheology measurement, the measurement chamber was cleaned before the next fluid sample entered the viscometer, and the process was repeated. The online viscometer measurements were in agreement with those from the API viscometers. The rheology data from the two instruments were well within the desired tolerance of ±1.5 dial readings at all rotational speeds investigated. This technology is combined with a density meter to form a density rheological unit (DRU). The DRU helps reduce nonproductive time and manage pressure drilling. However, the conventional mud tests still must be performed and recorded. This can verify the accuracy of the DRU measurement and provide redundant measurements in the event of a DRU failure [30].

### 3.2. Pipe Viscometer

Because the online rotational Couette viscometer is easily blocked, Vajargah [31], Magalhães [27], Sercan Gul [32], Knut Taugbøl [33,34], Hansen [2], Krogsæter [35], and Frøyland [36] used pipe viscometers to test drilling fluid rheological properties. According to Ahmed and Miska [37], the reliability and accuracy of pipe viscometers often outweigh rotational viscometers. Figure 4 shows a schematic example of the pipe viscometer. The test equipment requires a variable pump, a flow meter, mud with a known density, and a differential pressure sensor to measure the pressure difference in the test section of the straight pipe. Because screw pumps have no pressure pulsation and Coriolis flow meters can measure density and flow rate, pipe viscometers usually use a screw pump and a Coriolis flow meter. Pipe viscometer can measure drilling fluid rheological properties under laminar, transitional, and turbulent flow conditions. The data in the laminar flow state is calculated to characterize the rheological constant of the non-Newtonian fluid. The data obtained in transitional and turbulent conditions can be used to calculate the critical Reynolds number and friction factor in real time. Figure 5 shows the velocity profile in pipe laminar flow. The drilling fluid’s rheological properties can be obtained from the follow equations.

In laminar flow, wall shear stress can be obtained when the differential pressure across the measurement section is known:(5)$$\tau_{w}=\frac{D}{4}\frac{\Delta P}{\Delta L}$$
where *τ_w_* is the wall shear stress (Pa), *D* is pipe inner diameter (m), Δ*P* is the friction pressure loss (Pa), and Δ*L* is the pipe length of the test section (m).

It can be shown that for pipes, the shear rate at the wall can be obtained from [38]:(6)$${\dot{\gamma }}_{w}=\frac{1}{4}\left[3+\frac{d\left(ln\frac{8v}{D}\right)}{d\left(\tau_{w}\right)}\right]\left(\frac{8v}{D}\right)$$
where ${\dot{\gamma }}_{w}$ is the shear rate (1/s) and *v* is the velocity (m/s).

Introducing the generalized flow behavior index, *N*, as:(7)$$N=\frac{d(ln\tau_{w})}{d\left(ln\frac{8v}{D}\right)}$$

Then Equation (7) is now rewritten as:(8)$${\dot{\gamma }}_{w}=\left(\frac{3N+1}{4N}\right)\frac{8v}{D}$$

According to Equation (7), the slope of the “flow curve” (ln *τ_w_* vs. ln(8*v/D*)) represents the generalized flow behavior index, *N*. Once *N* is obtained from the flow curve, the shear rate at the wall can be calculated by using Equation (8). Subsequently, rheological parameters for any desired rheological model can be obtained by plotting the shear stress vs. shear rate at the wall and applying a proper curve fitting technique. The Herschel–Bulkley model (Equation (3)) exhibits an acceptable accuracy for the majority of drilling, completion, and cementing fluids and is therefore usually used to fit rheological curves.

Once the rheological constants are obtained, the Reynolds number can be calculated by mud velocity, density, and the wall shear stress as:(9)$$Re=\frac{8\rho v^{2}}{\tau_{w}}$$
where *ρ* is the mud density (kg/m^3^).

The friction factor *f* is calculated from the frictional pressure loss measurements using:(10)$$f=\frac{D}{2\rho v^{2}}$$

Vajargah et al. [31] designed and tested a pipe viscometer in 2016. The main measurement section was approximately 5.5 m long and consisted of two pipe sections 1.27 cm and 0.9525 cm in diameter. The pipe viscometer was calibrated with water before testing. Three types of drilling fluid (i.e., bentonite drilling fluid, polymer-based drilling fluid, and synthetic-based drilling fluid) were used to perform the pipeline rheometer. A standard viscometer was also used to obtain rheology data. The rheological diagrams of the two different methods are relatively similar. The test results are shown in Table 1.

The pipe viscometer designed by Sercan Gul et al. [32] has the test sections of the flow loop 1.25 m and 3.80 m long with an outside diameter of 2.54 cm. A comprehensive system calibration was achieved by circulating water at different flow rates through the flow loop. Excellent agreement was observed between the measurement results and the theoretical results. The mean absolute percentage error (MAPE) was calculated by taking the mean of the absolute percentage error (APE) for each single data point, as shown in Equations (11) and (12). The maximum APE was 3.5%, the MAPE was 1.6%, and the coefficient of determination (*R*^2^) was 0.99. In experimental verifications, a total of fifteen tests were performed using various water-based mud and oil-based mud formulations at 25 °C, 50 °C, and 65.5 °C. It showed the precision and robustness of the pipe viscometer method and that it could be used to provide a quality and frequent fluid characterization for field muds. In field testing, the pipe viscometer measurements of both PV and YP were a perfect match to the data reported in daily mud reports by the mud engineer.
(11)$$\mathrm{APE}=\left|\frac{\Delta p_{i-experimental}-\Delta p_{i-theoretical}}{\Delta p_{i-theoretical}}\right|*100\%$$
(12)$$\mathrm{MAPE}=\frac{1}{n}\overset{n}{\underset{i=1}{\sum }}|\frac{\Delta p_{i-experimental}-\Delta p_{i-theoretical}}{\Delta p_{theoretical}}|*100\%$$
where ∆*p* is the pressure loss (Pa), *i* is each measured data point, and *n* is the total number of measurements.

Taugbøl et al. [33,34] use the above principles to design pipe automatic drilling fluid measurement. The total length of the equipment was 3.3 m, the width was 0.7 m, and the height was 0.9 m. It converted the measurement results of the pipe viscometer to the standard Couette viscometer’s dial readings and achieved good results. It was set to measure the viscosity every 15 min. It was installed and ran on several offshore drilling platforms and provided high-quality real-time drilling fluid data. It highly reduced the sampling time intervals, enabling improved fluid control and improved fluid quality.

Magalhães Filho et al. [27] compared three drilling fluid rheological property real-time testing methods: a standard Couette viscometer FANN35A, online Couette viscometer TT-100, and pipe viscometer. For the Newtonian fluids, the viscosities measured by the three instruments were consistent. For non-Newtonian fluids, the PL parameters provided by each instrument were different, including drilling fluid (with suspended solids) and polymer solutions. These variations can mainly be caused by the consequences of fluid/gap interfaces, homogeneity, and slipperiness. However, these differences were not severe when the pressure drop was estimated using these parameters. The error between the pressure drop calculated by the TT-100 and the experimental value was the smallest. The error of the FANN35A was 17.81%, the error of the TT-100 was 0.41%, and the error of the pipe viscometer was 6.27%.

Some methods have the same theory as the pipe viscometer. Compared with the abovementioned pipe viscometers, these methods are convenient and do not take up much space. A novel downhole sensor was developed by Rondon et al. [39]. It can be inserted into the drill string to measure the rheological properties of fluids in real-time. The mixtures of glycerin and water were used to test and calibrate this sensor. Real crude oil samples were also used to test the performance of this sensor. The error between the designed sensor’s measurement value and the standard measurement value was within 2%. However, drilling fluids need further testing to evaluate the performance of the sensor. This sensor needs to further consider the flow rate and viscosity range of the drilling fluid and optimize the dimensions of the sensor. Carlsen et al. [40] measured the pressure at various positions in the drilling fluid’s circulation system. Various flow rates and pressures were used to measure the friction coefficient of the drilling fluid. The results show that it can also be used to calculate other drilling fluid rheological properties such as shear stress and viscosity. Vajargah et al. [41,42] proposed a method to determine rheology in real time from downhole measurements of pressure drop and temperature, considering the well as an annulus pipe viscometer. It can directly obtain the ECD of the well. The results were compared to offline data taken from an offline high-pressure, high-temperature rheometer. It can estimate the gel strength by peak pressure loss. However, the time-dependent behavior of the drilling fluid theory needs to be developed through this method. We think, with the development of measurement while drilling (MWD) technology, it is a good method to obtain the rheological properties of drilling fluid by measuring downhole pressure. This method does not require further surface measurements, which can greatly simplify the rheological measurement methods and equipment and eliminate the labor required for operation and maintenance.

Pipe viscometers usually use a round pipe, the sensor installed in the round pipe may affect the flow rate, resulting in inaccurate pressure measurement. Therefore, Liu et al. [43] developed a rectangle pipe viscometer. During the test, a 5% bentonite slurry with a density of 1.03 g/cm^3^ was prepared, and 0.1% polyacrylamide glue solution, 0.1% polyacrylonitrile ammonium, and xanthan gum was added successively. The pipe viscometer continuously recorded the change process of the rheological properties of the drilling fluid, the measurement results were the same as the standard viscometer measurement results, and the error was small. Sun et al. [44] developed a type of altered-diameter rectangle pipe viscometer to realize continuous online monitoring. Through altered-diameter pipes, different flow rates can be generated under constant flow. Fresh water and bentonite drilling fluid are tested in the laboratory. Compared with the standard viscometer, the results show that the error of AV and PV are both within the allowable range. The field test results show that the performance of the tested data was stable and reliable. Compared with the API standard method, the error was within the allowable range.

While a pipe viscometer takes up a lot of space, Sercan Gul [45,46] developed a helical pipe viscometer to measure rheological properties. The system included four test parts (two horizontal straight pipes and two vertical spiral pipes), four differential pressure sensors, a 40 L liquid storage tank, a Coriolis flowmeter, and a variable frequency drive screw pump. It tested the rheological properties of 20 polymer-based fluids. These fluids were based on water and added xanthan gum to increase the viscosity. The equivalent straight pipe pressure losses needed to be calculated accurately by using the pressure loss data obtained from a helical pipe viscometer. However, none of the papers [47,48,49] reported correlations of PL fluids were valid for Herschel–Bulkley fluids. Thus, a random forest regression model was used to predict the friction coefficient with friction pressure loss, the mean absolute error was 0.803 × 10^−3^, and the mean absolute percentage error was 4.55%. The algorithm was developed using the trained machine learning model and the pipe viscometer equations. The rheogram results matched the standard Couette viscometer.

Table 1 shows a comparative study of the online Couette viscometers and the pipe viscometers conducted by previous researchers, and the results are similar to the standard Couette viscometer.

### 3.3. The Technology Based on Marsh Funnel

Marsh funnel is a commonly used instrument for viscosity analysis, calculating the final fluid release time of almost 1.5 L. Marsh [50] invented the Marsh funnel (Figure 6) in 1931 as a quick and easy way to estimate the viscosity of drilling fluids. Pitt et al. [19,51,52,53,54,55,56] analyzed the relationship between rheological properties and the Marsh funnel and established models for estimating drilling fluid rheological properties by using Marsh funnel time such as Equation (13) [51], Equation (14) [53], and Equation (15) [53]:(13)AV=ρ(t−25)
(14)$$\mathrm{AV}=-0.0118*t^{2}+1.6175*t-32.168$$
(15)AV=ρ(t−28)
where *t* is March funnel time, seconds.

An artificial neural network (ANN) is an effective technology that imitates the biological neurons of the human brain [57]. The main processing element of the artificial neural network system is the neuron. The ANN model is composed of network architecture with at least three layers (i.e., input, hidden, and output layers), a training algorithm, and a transfer function [58]. Each layer is connected to other layers, and the constants of these layers are called weights. The backpropagation technology is used in the training of artificial neural networks. By Comparing the estimated data and actual data in the output layer, it updates the weight between the deviation of each connection and each layer. This is repeated until the desired improvement is achieved and the error is reduced to a certain threshold [59].

According to the above theories, artificial intelligence methods are used to estimate rheological properties more accurately in real time based on parameters such as the Marsh funnel viscosity, mud weight, and solid content [60,61,62,63,64,65,66,67,68,69,70,71]. Bispo et al. [72] used temperature, xanthan gum, bentonite, and barite to estimate AV. The main model used was the ANN. Figure 7 shows a schematic of the ANN model to estimate rheological properties. The model usually consists of three layers: an input layer—in addition to the Marsh funnel viscosity, the other drilling fluid parameters, such as mud weight and solid content, are also added as input elements; a hidden layer, which contains an optimized number of neurons; an output layer, which contains output parameters (PV, YP, AV, *K*, *n,* and *τ*_0_).

Abdelgawad et al. [63] and Elkatatny et al. [66] used the self-adaptive differential evolution (SaDe) algorithm to optimize the best combination of the ANN’s parameters for rheological property estimation. SaDe proposed by Qin et al. [57] is a special differential evolution algorithm with adaptive control parameters and mutation strategies based on learning experience [57,58]. The trial-and-error procedure required to obtain the optimized solution is avoided in SaDe, which reduces the time required for optimization problems [70].

As shown in Table 2, scholars use the ANN model to predict the drilling fluid rheological properties (PV, AV, *n*, *K*, *τ*_0_, reading of 300 rpm and 600 rpm) in real time and show good results. The methods to evaluate the performance of the ANN model are average absolute error (AAE) (Equation (16)), average absolute percentage error, AAPE (Equation (17)), *R*^2^, and mean square error (MSE). In all these papers, the predicted result for *R*^2^ was greater than 0.89, AAE was less than 4.7, and AAPE was less than 8.6%. This inexpensive technique will help drilling engineers to control the drilling operation better and predict drilling problems before they occur. Moreover, it will reduce the total cost of drilling operations. However, different ANN models need to be established by training drilling fluid data in different mud systems. One artificial intelligence model can be used in wells located in the same block or in the same drilling fluid system.
(16)$$\mathrm{AAPE}=\frac{1}{n}\sum_{i=1}^{n}|\left(\frac{y_{real}-y_{predict}}{y_{real}}\right)*\;100|$$
(17)$$\mathrm{AAE}=\frac{1}{n}\sum_{i=1}^{n}|y_{real}-y_{predict}|$$
where *i* is each measured data point, *y_real_* is real measured value, and *y_predict_* is predict measured value.

### 3.4. Acoustic Technology

In 2011, Miller et al. [73] introduced a system for continuously measuring and recording mud density and viscosity, using an instrument based on tuning fork technology as shown in Figure 8. The instrument can measure density and viscosity, and the viscosity is in units of equivalent Marsh funnel seconds. The instrument was mounted in-line with a constant fluid flow rate past it. This method has a wide spectrum of industrial applications; it is robust and reliable and takes into account the separation of the fork teeth, so it is not easy to block. The instrument measures the standing wave generated by the vibrating teeth of the fork and calculates the density and viscosity of the fluid based on the amplitude and frequency measurement results. The density and viscosity values are output with 4–20 mA signals, which can be displayed locally in any required oilfield unit and output to the drilling rig data collection or logging tool computer software package.

The tuning fork instrument kit operates at a back pressure of 0.7–0.8 MPa, so the density read is equal to the density of the normal pressurized drilling fluid. Density and viscosity are calibrated by manual readings per shift. The instrument kit can be recalibrated based on the new readings of the standard Marsh funnel viscometer when the density or viscosity significantly changes. Experience in fields showed that a calibration check is sufficient at the start of each shift to ensure that readings are comparable directly to manual readings. The sensor is small and easy to install but requires manual calibration.

Ofoche et al. [74] presents a novel approach of continuously measuring drilling fluid rheology and density by use of sound signals. Figure 9 shows the schematic diagram. The flow of drilling fluid in the pipeline is Poiseuille flow. Sound waves generated by a piezoelectric transducer are passed though the fluid and the resultant damping effect of the signals are used to drive a receiver piezo disc. The data acquisition device records the frequency response and voltage by use of a fast Fourier transform routine. A flow loop with a constant rate designed to simultaneously calculate the six API recommended shear rates. Therefore, six pipe sections were designed that have diameters corresponding to the six normal shear rates used in the petroleum industry. Since both density and viscosity will affect the signal response, a multivariate random forest method was established and used to predict the dial readings. All dial readings measured by the acoustic method were within ±1 of the dial readings of the rotational viscometer.

## 4. Discussion and Prospects

Table 3 briefly compares the real-time measurement techniques, and the detailed analysis are as follows.

(1)The online Couette viscometer is the most similar to the API standard measurement method, so it has the highest accuracy and can measure all drilling fluid rheological properties. However, the gap between the rotor and the stator is narrow, and the diameter of solids must less than 1 mm. Solid or gels particles may be sedimented in the viscometer, so the online Couette viscometer is easily plugged. It is inconvenient to use and requires regular cleaning and maintenance. This viscometer is suitable for drilling fluids with low viscosity and low solid content.(2)Compared with the online Couette viscometer, the pipe viscometer provides better automatic measurement technology. The solid and gel particles in the drilling fluid will not settle in the pipe. By adding additional sensors to the pipe, additional variables can be obtained such as fluid density, temperature, critical Reynolds number, and real-time friction coefficient. However, it cannot measure the 10 s and 10 min gel strength. The pipe viscometer is large, and it occupies a large space for installation. Compared with the pipe viscometer, the helical pipe viscometer has obvious advantages, having a compact size and more general friction pressure loss curve. At the same time, the helical pipe increases the friction pressure loss and delayed flow state transition, so the helical pipe viscometer can be used to collect more data in the laminar flow state, thereby improving the accuracy of low shear rheological parameter estimation. However, the theoretical basis for the helical pipe viscometer is still under development.(3)Artificial intelligence technology is the cheapest method, because only the Marsh funnel is needed, and the mud balance and solid content meter may be added optionally. Although the test of the Marsh funnel, density, and sand content was simple and quick, it still required manual testing. The neural network model was different when the drilling system was different. One artificial intelligence method can be used in wells which are in the same block or in the same drilling system. The test results of the tuning fork technology were Marsh funnel viscosity and density, which can be combined with artificial intelligence technology to form an automatic online measurement of drilling fluid rheological properties.(4)The current drilling fluid rheology measurement is offline in API recommended practice for field testing drilling fluids, which can no longer meet the needs of intelligent drilling. In order to control and optimize rheological parameters in real-time, it is necessary to consider developing a criterion that can measure drilling rheological properties in real time. The standard Couette viscometer also has shortcomings [75,76,77,78] analyzing the end-effect, correction, and reliability of the Couette viscometer. It is an opportunity to use real-time drilling fluid rheological properties measurement to improve the current criteria. The pipe viscometer is good in automation and will not be plugged by solids and gel. The reliability and accuracy of pipe viscometers often outweigh rotational viscometers. At present, the pipe viscometer used is large, so it is necessary to miniaturize the instrument for convenient use in the field. The helical pipe viscometer requires further mechanism research in theory to promote the real-time measurement of rheology.(5)The practicability of the instrument is relatively not good. We think two convenient methods can be considered in the future. One method is using MWD technology to measure the pressure in the drill string and annulus while drilling, the ECD can be accurately obtained. The drill pipe and annulus are considered as a large pipe viscometer, which can calculate the drilling fluid rheological properties. The other method is acoustic technology, including tuning fork technology, ultrasonic technology, etc. There are many articles using ultrasound to measure fluid parameters [79,80,81,82,83], but the composition of the drilling fluid is complex, and the ultrasonic attenuation is related to many factors [26] (temperature, density, viscosity, solid content, etc.). For example, if an increase in ultrasonic attenuation is due to the entrance of solids into the system, the density should also increase, and some increase may be observed in the viscosity. On the other hand, if an increase in the attenuation is observed due to the addition of polymers, the density may change slightly or not even change, but viscosity will significantly increase. The sound speed is also important since it helps the system to discern when the density is rising due to the solids suspended or solids dissolved. Therefore, the theory of ultrasound technology needs to be developed by simulation and experiment [21,22,83]. Magalhães et al. [26] used density, viscosity, ultrasound attenuation, and sound speed as inputs to establish an ANN model of concentration of the suspended solids. The installation of these two methods is convenient, but further technology and theoretical research are needed.(6)The current drilling fluid performance testing mainly measures the drilling fluid that samples from the mud pit. The drilling fluid returning from the annulus contains a lot of stratum information, so its testing is also very important and can help judge the formation. Testing its performance can also make better decisions for processing to maintain the performance of drilling fluid. However, the mud returning from the annulus contains many solid particles, and some particles have large diameters. Therefore, the allowable particle diameter of the instrument needs to be further considered in the selection of equipment.

## 5. Conclusions

This article analyzes the four real-time measurement technologies of drilling fluids, and the measurement results of each technology are within the tolerance range compared with the standard rotational Couette viscometer in field test. At present, there is no industry standard handbook for real-time drilling fluid measurement. This status hinders the automatic control of drilling fluid. In the future, technical standards for real-time drilling fluid measurement will be established based on standard laboratory testing methods. This article analyzed the four methods in terms of principles, implementation methods, advantages, limitations, etc. Engineers can choose one of the four methods for real-time measurement according to their requirements in the field. The online rotational Couette viscometer is suitable for the low-viscosity and low-solid content drilling fluid. The pipe viscometer is a reliable real-time measurement method. As a result, a standard and small pipe viscometer may be formed to test the drilling fluid rheological properties in the future. The technology of MWD downhole pressure measurement and ultrasound technology can also be considered to measure drilling fluid rheological properties. Through real-time measurement of the rheological properties of drilling fluids, the performance of drilling fluids can be grasped in real-time, the drilling status can be judged in time, and relevant drilling fluid processing decisions can be made to ensure drilling safety. Through big data, artificial intelligence, and other technologies, the drilling fluid performance can be optimized in real time to achieve the optimal rate of penetration, thereby improving economic benefits.

## Figures and Tables

**Figure 1 sensors-21-03592-f001:**
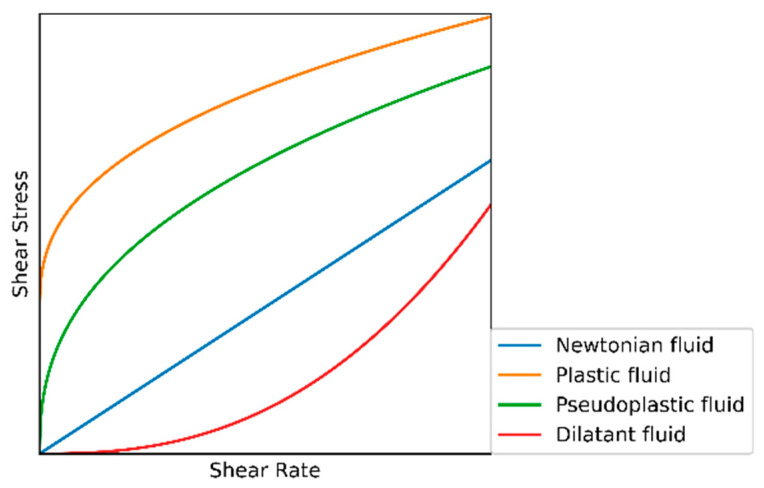
Rheological curves of Newtonian fluid, plastic fluid, pseudoplastic fluid, and dilatant fluid.

**Figure 2 sensors-21-03592-f002:**
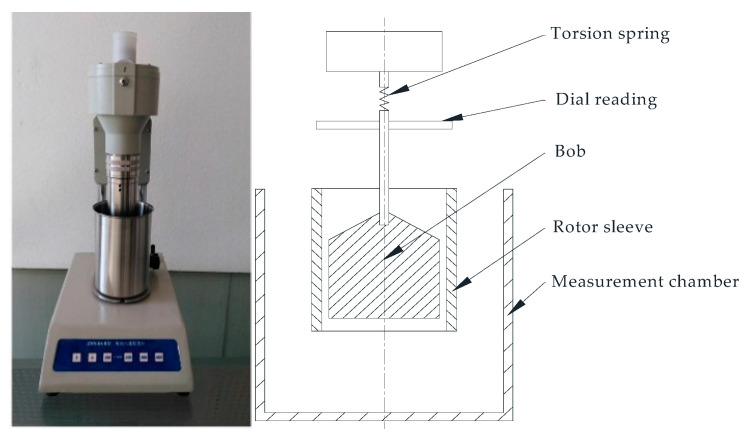
Standard rotational Couette viscometer.

**Figure 3 sensors-21-03592-f003:**
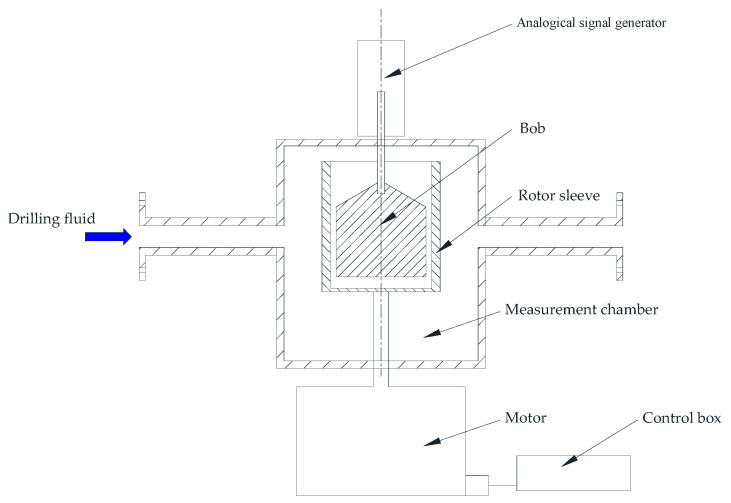
A typical online rotational Couette viscometer schematic diagram.

**Figure 4 sensors-21-03592-f004:**
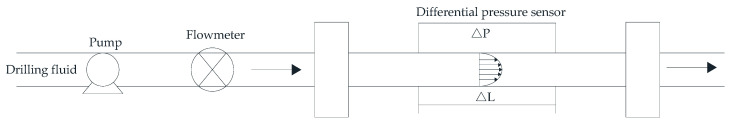
A schematic example of the pipe viscometer.

**Figure 5 sensors-21-03592-f005:**
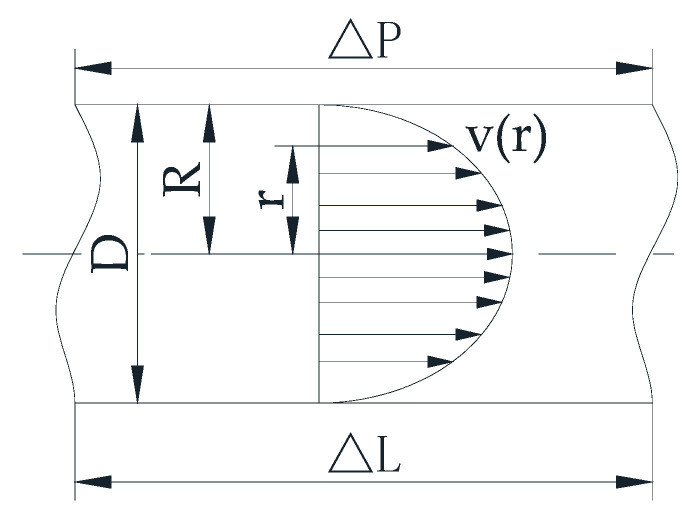
The velocity profile in pipe laminar flow.

**Figure 6 sensors-21-03592-f006:**
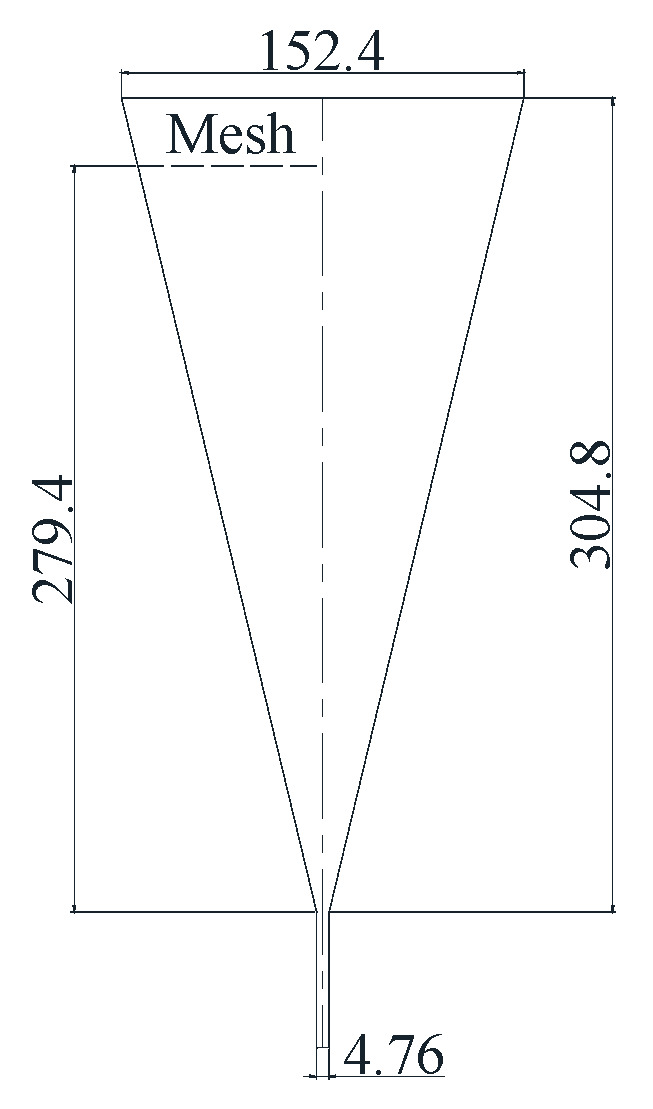
Marsh funnel.

**Figure 7 sensors-21-03592-f007:**
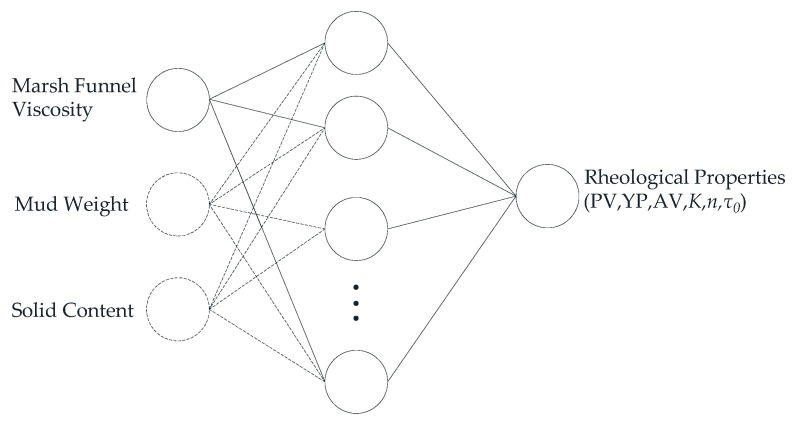
The ANN model for rheological properties estimation.

**Figure 8 sensors-21-03592-f008:**
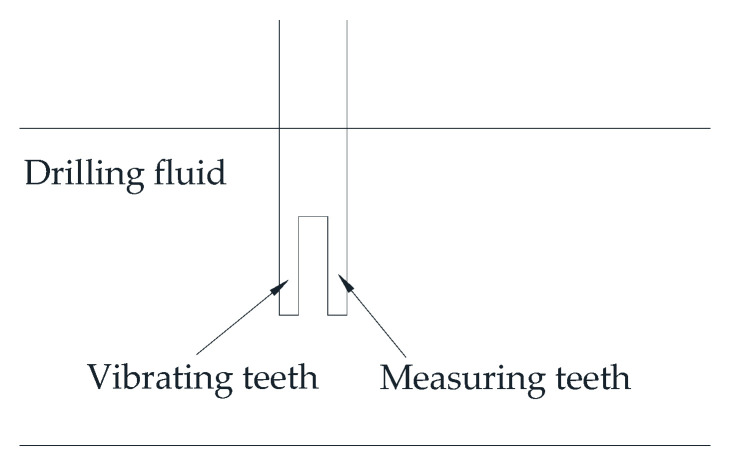
Tuning fork technology.

**Figure 9 sensors-21-03592-f009:**
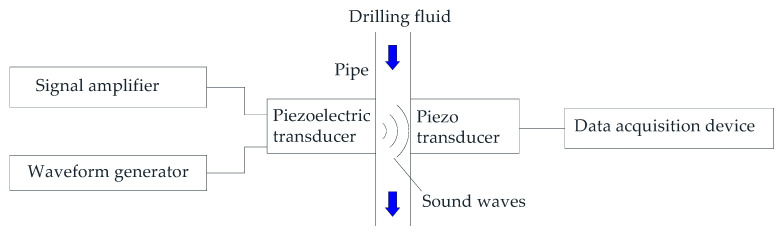
Schematic diagram in Ofoche et al. [74].

**Table 1 sensors-21-03592-t001:** Comparative studies of online Couette viscometers and pipe viscometers conducted by previous researchers.

Researcher	Measurement Technology	Drilling Fluid	Fluid Type	Temperature (°C)	Performance Criteria										Difference between Real-Time and Standard Viscometer			
Magalhães et al. [26]	TT-100 (online Couette viscometer)	Glycerin	Newtonian	32	*μ*										Error (%)			
					TT-100				FANN 35A						3.16			
					16.3				15.8									
		CMC solution	Non-Newtonian	33	*K*				*n*						MSE	R^2^		
					TT-100		FANN 35A		TT-100		FANN 35A				3.9365	0.9928		
					2.72		3.24		0.46		0.44							
		Water-based mud	Non-Newtonian	34	*K*				*n*						MSE	R^2^		
					TT-100		FANN 35A		TT-100		FANN 35A				20.7916	0.9383		
					1.85		4.20		0.48		0.37							
		Non-aqueous drilling fluid	non-Newtonian	51	*K*				*n*						MSE	R^2^		
					TT-100		FANN 35A		TT-100		FANN 35A				20.9649	0.9697		
					0.07		0.17		1.00		0.85							
Dotson et al. [29]	Online Couette viscometer	Oil-based drilling fluid	Non-Newtonian	-	Mean difference; 95% confidence interval; Standard deviation between Real-time and standard viscometer (dial readings)										-			
					600 rev/min		300 rev/min		100 rev/min		6 rev/min							
					0.6; [0.3,0.9]; 1		0.5; [0.3,0.7]; 0.7		−0.1; [−0.2,0], 0.4;		−0.5; [−0.7,−0.4]; 0.4							
Vajargah et al. [31]	Pipe Viscometer	Bentonite clay suspensions	Non-Newtonian	-	*K*			*n*			*τ* _0_				MSE	R^2^		
					Couette		Pipe	Couette	Pipe		Couette		Pipe		1.9234	0.9362		
					0.06292		0.03438	0.789	0.8879		8.001		6.525					
		Polymer-based	Non-Newtonian	-	*K*			*n*			*τ* _0_				MSE	R^2^		
					Couette		Pipe	Couette	Pipe		Couette		Pipe		0.6342	0.9976		
					1.900		2.430	0.4591	0.4197		0		0					
		Synthetic-based drilling fluid	Non-Newtonian	-	*K*			*n*			*τ* _0_				MSE	R^2^		
					Couette		Pipe	Couette	Pipe		Couette		Pipe		1.5635	0.9943		
					0.1284		0.1753	0.8456	0.7912		1.736		2.924					
Gul et al. [32]	Pipe Viscometer	Oil- based mud	Non-Newtonian	25	*K*		*n*		*τ* _0_		PV		YP		MSE	R^2^		
					Ofite 900	Pipe	Ofite 900	Pipe	Ofite 900	Pipe	Ofite 900	Pipe	Ofite 900	Pipe	2.5967	0.9939		
					0.31	0.28	0.75	0.77	0.51	1.29	45	46	9.5	10				
		Oil- based mud	Non-Newtonian	65.5	*K*		*n*		*τ* _0_		PV		YP		MSE	R^2^		
					Ofite 900	Pipe	Ofite 900	Pipe	Ofite 900	Pipe	Ofite 900	Pipe	Ofite 900	Pipe	3.6923	0.9522		
					0.10	0.14	0.79	0.76	0.16	0.14	21	22	5.3	4.8				
Magalhães et al. [27]	Online Couette viscometer; Pipe viscometer	Glycerin 50%	Newtonian	32	*μ*										Error (%)			
					TT-100			FANN 35A				Pipe			TT-100	Pipe		
					16.3			15.8				15.5			3.16	1.90		
		Glycerin 50%	Newtonian	50	*μ*										Error (%)			
					TT-100			FANN 35A				Pipe			TT-100	Pipe		
					9.1			8.6				8.2			5.81	4.65		
		0.25% CMC	Non-Newtonian	30	*K*				*n*						MSE		R^2^	
					TT-100		FANN 35A	Pipe	TT-100			FANN 35A	Pipe		TT-100	Pipe	TT-100	Pipe
					0.10		0.40	0.43	0.75			0.52	0.52		4.845	0.441	0.827	0.984
		0.25% CMC	Non-Newtonian	50	*K*				*n*						MSE		R^2^	
					TT-100		FANN 35A	Pipe	TT-100			FANN 35A	Pipe		TT-100	Pipe	TT-100	Pipe
					0.03		0.24	0.16	0.88			0.56	0.60		1.09	0.866	0.939	0.952
		1% CMC	Non-Newtonian	33	*K*				*n*						MSE		R^2^	
					TT-100		FANN 35A	Pipe	TT-100			FANN 35A	Pipe		TT-100	Pipe	TT-100	Pipe
					2.72		3.24	3.96	0.46			0.44	0.4		3.937	6.966	0.993	0.988
		1% CMC	Non-Newtonian	50	*K*				*n*						MSE		R^2^	
					TT-100		FANN 35A	Pipe	TT-100			FANN 35A	Pipe		TT-100	Pipe	TT-100	Pipe
					1.19		1.66	2.08	0.55			0.50	0.46		0.886	1.788	0.997	0.995
		Water-based drilling fluid	Non-Newtonian	34	*K*				*n*						MSE		R^2^	
					TT-100		FANN 35A	Pipe	TT-100			FANN 35A	Pipe		TT-100	Pipe	TT-100	Pipe
					1.85		4.2	3.55	0.48			0.37	0.38		20.79	12.27	0.939	0.964
		Water-based drilling fluid	Non-Newtonian	50	*K*				*n*						MSE		R^2^	
					TT-100		FANN 35A	Pipe	TT-100			FANN 35A	Pipe		TT-100	Pipe	TT-100	Pipe
					1.34		3.15	2.60	0.50			0.39	0.40		20.72	13.07	0.920	0.950
Baoshuang et al. [43]	Pipe viscometer	Water-based drilling fluid	Non-Newtonian	-	AV				PV						Error (%)			
					API			Pipe	API				Pipe		AV		PV	
					9.50			9.41	4.60				4.82		0.947		4.782	
Haoyu et al. [44]	altered-diameter shaped pipe viscometer	Drilling fluid	Non-Newtonian	-	AV		PV		YP		*n*		*K*		Error (%)			
					API	Pipe	API	Pipe	API	Pipe	API	Pipe	API	Pipe	AV	PV	YP	
					12.5	12.37	9	8.85	3.5	3.47	0.64	0.63	0.2	0.01	4.783	1.67	0.857	

*μ,* dynamic viscosity; *K*, consistency factor; *n,* flow behavior index; *τ*_0_*,* fluid yield stress; AV, apparent viscosity; PV, plastic viscosity; YP, yield point; MSE, mean square error.

**Table 2 sensors-21-03592-t002:** Comparative studies of various AI techniques conducted by previous researchers.

Researcher	Drilling Fluid	Input Parameters	AI Technique	Number of Data Points	Performance Criteria																	
Elkatatny et al. [60]	NaCl polymer mud	Marsh funnel viscosity, solid content; mud weight	ANN	3000	300		600		PV		AV			*τ* _0_			*n*			*K*		
					R^2^	AAE	R^2^	AAE	R^2^	AAE	R^2^	AAE		R^2^		AAE	R^2^	AAE		R^2^		AAE
					0.974	3.27	0.978	3.51	0.977	4.7	0.9792	3.4		0.8998		3.67	0.9487	2.1		0.8865		0.89
Elkatatny et al. [61,62]	invert emulsion-based mud	Marsh funnel viscosity, solid content; mud weight	ANN	9000	300		600		*n*		*K*			AV			PV			*τ* _0_		
					R^2^	AAE	R^2^	AAE	R^2^	AAE	R^2^	AAE		R^2^		AAE	R^2^	AAE		AAE		
					0.8981	3.48	0.9235	3.7	0.954	1.2	0.9205	4.7		0.9235		3.7	0.9452	4.2		3.0		
Abdelgawad et al. [63]	-	Marsh funnel viscosity, solid content; mud weight	SaDe-ANN	2000	AV			PV				*τ* _0_					*n*					
					R^2^		AAPE	R^2^		AAPE		R^2^		AAPE			R^2^				AAPE	
					0.945		5.39%	0.94		3.91%		0.928		4.71%			0.922				3.26%	
Elkatatny et al. [64]	NaCl polymer mud	Marsh funnel viscosity, solid content; mud weight	ANN	1000	300		600		*n*			*K*		PV					AV			
					R^2^	AAPE	R^2^	AAE	R^2^	AAE		R^2^	AAE	R^2^			AAE		R^2^		AAE	
					0.99	3.46%	0.99	3.43%	0.96	3.25%		0.92	6.50%	0.98			6.00%		0.99		3.96%	
Al-Khdheeawi et al. [65]	Ferro Chrome Lignosulfonate mud; Salt Saturated mud	Marsh funnel viscosity, mud weight	ANN	142	AV																	
					R^2^							AAE										
					0.981							0.109										
Elkatatny et al. [66]	NaCl polymer mud	Marsh funnel viscosity, solid content; mud weight	SaDe-ANN	900	PV		*τ* _0_			*n*					*K*		AV					
					R^2^	AAPE	R^2^	AAPE		R^2^		AAPE			R^2^	AAPE	R^2^				AAPE	
					0.96	8.60%	0.95	3.50%		0.94		4.00%			0.91	8.40%	0.96				5.80%	
Gowida et al. [67]	CaCl_2_ Brine-Based	Marsh funnel viscosity, mud weight	ANN	515	PV		YP			AV					*n*		*K*					
					R	AAPE	R	AAPE		R		AAPE			R	AAPE	R				AAPE	
					0.98	6.1	0.97	3.9		0.99		3.2			0.98	2.4	0.99				3.6	
Alsabaa et al. [68,69]	Oil-based mud	Marsh funnel viscosity, mud weight	ANN	369	PV		YP		*n*			AV		300					600			
					R	AAPE	R	AAPE	R	AAPE		R	AAPE	R			AAPE		R		AAPE	
					0.95	7.97	0.9	6.03	0.91	4.81		0.94	6.9	0.92			6.74		0.94		6.95	
Alsabaa et al. [70]	Invert emulsion mud	Marsh funnel viscosity, mud weight	ANFIS	741	PV		YP		*n*			AV		300					600			
					R	AAPE	R	AAPE	R	AAPE		R	AAPE	R			AAPE		R		AAPE	
					0.91	5.66	0.91	3.38	0.94	1.69		0.97	2.59	0.93			3.47		0.97		2.59	
Gomaa et al. [71]	High-overbalanced water-based mud	Marsh funnel viscosity, mud weight	ANN	3000	PV		YP		*n*			AV		300					600			
					R	AAPE	R	AAPE	R	AAPE		R	AAPE	R			AAPE		R		AAPE	
					0.94	7.7	0.91	3.03	0.94	2.5		0.96	3.96	0.97			3.7		0.96		4.77	
Bispo et al. [72]	Water-based mud	temperature xanthan gum, bentonite and barite	ANN	1017	AV																	
					R^2^							MSE										
					0.9486							7.73										

*μ,* dynamic viscosity; *K,* consistency factor; *n,* flow behavior index; *τ*_0_*,* fluid yield stress; AV, apparent viscosity; PV, plastic viscosity; YP, yield point; 300, reading of 300 rpm; 600, reading of 600 rpm; AAE, average absolute error; AAPE, average absolute percentage error; MSE, mean square error.

**Table 3 sensors-21-03592-t003:** Comparison of measurement techniques.

Technique	Working Principle	Advantages	Limitations	Cost	Notable Reference
Online Couette viscometer	Concentric cylinder (Couette flow)	Similar to API standards	Solids less than 1 mm; solids settling; easily blocked; frequent maintenance	High	[21,22,23,24,25,26,27,29]
Pipe viscometer	Pipe pressure difference under various flow rates	Automation; not susceptible to blockage; obtains other parameters by adding other sensors to the pipe	Large size	High	[2,27,31,32,33,35,36]
Based on Marsh funnel	Marsh funnel time, mud weight, solid content	Simple test tool	Manual test, complex theoretical model	Low	[19,51,63,64,65,66,67,68,52,53,54,55,56,60,61,62]
Acoustic technology	Acoustic characteristics of sound waves propagating in drilling fluid	Simple installation; not susceptible to blockage; density and viscosity can be measured	Manual calibration; complex theoretical model	Medium	[73,74]

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
