# Peer review of "Real-Time Measurement of Drilling Fluid Rheological Properties: A Review"

_sensors, 2021, doi:10.3390/s21113592_

Round 1

Reviewer 1 Report

Dear authors, this review paper presents interesting comparison of real time drilling fluid rheology measurement techniques. The paper is easy to read and would be useful to the readers. However, I have several comments/suggestions to improve the writing quality, clarity, and technical presentation of the manuscript. Below are my comments: 

  1. A table providing brief comparison of all measurement techniques would be a useful addition to the manuscript. Although you have discussed it in the text, tabular format summarizing the main contrasts in terms of working principle, advantages, limitations, costs, notable references, etc. would be very useful for cursory readers who want to quickly get overview of different approaches.
  2. The introduction section is very short for a review paper. You should add more discussion and information to highlight important limitation of current industry practices and how real time monitoring of drilling fluids rheology can be very useful. You can perhaps add some statistics related to drilling fluid costs, no of incidents caused by lack of control over drilling fluid rheology, examples from field implementation, etc. to make a strong case in favor of real time drilling fluids measurements.
  3. References – There have been several publications on this topic particularly in 2020 and 2021. You can find them on onepetro and google scholar. Please review and discuss them as well so that latest developments are also incorporated in this paper.
  4. Although you have explained the working principle and theory behind each technique, please provide more details on system/infrastructure (sensors, equipment, etc.) needed for real time drilling fluids monitoring using each of the system preferably with some images or labeled schematic. This is important as this journal focuses on sensors. You can also discuss some case studies and costs/economic considerations of real time monitoring system. This will provide more holistic comparison of all the techniques.
  5. Other techniques – Although you have mentioned ultrasonic method very briefly, it is worth discussing other techniques such as ultrasonic, electro-chemical/mechanical, nuclear NMR, etc. It will make the manuscript more rounded. You can combine all these techniques under a miscellaneous category and perhaps explain their biggest limitations and why they are not popular yet. Although you are focusing on rheology, brief discussion on density, cuttings, and other real time measurements that could be relevant to rheology may also be useful.
  6. Table 1 – It is not clear what do you mean by the column “Performance Criteria”. For second row (Reference [45]), it is clear that the numbers presented indicate deviation or difference between real time measurement and actual viscometer measurement. What about other rows? What do K, n value shown mean? I suggest that instead of listing K and n values, you calculate the % difference between the viscometer and real time measurement and present in this table. Percentage differences will make comparisons among techniques easier.
  7. Minor comments:
    • Figure 1 – Instead of just symbols, provide text description in axis labels such as “Shear Stress” and “Shear Rate”
    • Table 1 and 2 – Along with the reference number, please add researcher name. For example, “Vajargah et al. [28]”
    • Table 2 - Mention full form of AV, PV, AAE, and other symbols as a footnote in the table itself. It will be more convenient for readers.
    • English/formatting revisions:
      • Line 124 – “scholars”
      • Although ‘on-line’ is not wrong, “online” is generally preferred
      • In-text citations should be in consistent format. E.g. Vajargah A K 2016[28] should be Valargah et al. [28]. Please review and fix all.
      • Please review English and revise/rephrase sentences for correct grammar or better clarity. Please be specific in your statements. Following are just some examples, please review the paper and revise wherever necessary.
        • Line 13 – “It is also important for intelligent drilling, providing drilling fluid data to establish an optimization model (of what?), optimizing drilling process parameters (?)”
        • Line 24 – “Via real-time measurement of the drilling fluid rheological properties, real-time optimization of the drilling fluid (redundant) and judgment of the drilling status (ROP?) can be achieved”
        • Line 56 – “The real-time measurement of drilling fluid can also promote the automation process of drilling fluid (Design? Control?) and realize(?) the closed-loop control of drilling fluid
        • Line 401 - “But the theory of helical pipe viscometer is still needing….” can be rephrased as “.. still needs development” or “..is still under development”

Reviewer 2 Report

This manuscript reviews different real-time measurement techniques of drilling fluid rheological properties. The authors have attempted to summarise techniques of a number of studies conducted; thus, the manuscript is rich in technical information. However, often I see outlines of each different studies than a cohesive critical review; thus, overall, the manuscript has failed to develop organization and transitions within the writing and draw meaningful conclusions. A major revision is required before publication.

  1. The abstract needs to be re-written and perfected. The current version mainly points outs the importance of obtaining real-time rheological parameters than the contribution of the study.
  2. Line 23: Sentences, i.e. “This paper provides references...” devalues the contribution.
  3. Line 25-26: The sentence is not clear.
  4. The introduction section is failed to provide background information on current drilling technology, different drilling fluids and available technologies/ standards to determine rheological properties. The overall flow of the paper, particularly the introduction section, is poor. Thus, repetition is often (i.e. the importance of real-time measurement fluid rheological properties is discussed on and on)
  5. Line 34-35: Sentence is not meaningful “the right drilling fluid program… will have serious consequences”. It is strongly suggested to proofread the manuscript and check it is readability.
  6. Line 64: Equivalent->equivalent
  7. Line 78: Provide examples for different fluid types shown in Fig.1. How are rheological curves critical to characterize drilling fluids?
  8. What are τ and γ in Figure 1?
  9. Line 97-98: Sentence is unclear
  10. Line 113: “device. And” check
  11. Line 129-175: These four paragraphs summarise several independent studies that employed rotational Couette viscometers. Unless a reader checks technical details of several Couette viscometer tests, this technical information is not meaningful. It is recommended to highlight the uniqueness of each technique, compare, contrast and critically evaluate them. Technical information can be presented in a Table for better comparison.
  12. Line 181: Explain the principle of the pipe viscometer. Fig. 3 has no meaning.
  13. Line 195: pip -> pipe
  14. Make sure to introduce all the symbols used in the equations.
  15. Include a schematic diagram of a pipe rheometer and explain.
  16. It could have been interesting to see the schematic diagrams of selected unique test techniques to compare and contrast.
  17. The performance criteria of Table 1 and 2 are not consistent can clear. Consider adding additional columns for clarity.
  18. Line 333: Bring the introductions of Artificial Neural Network before explaining how those can be used to determine real-time based on parameters (i.e. Line 324)
  19. Explain Fig. 5.
  20. Showcase a tuning fork instrument (schematic). Why it is “strong and reliable” (Line 369)
  21. Line 402: Justify why the “artificial intelligence technology is the lowest cost method”
  22. Line 423: “The current online measure system is large” not meaningful

Reviewer 3 Report

Dear authors,

This paper provides four real-time measurements of drilling fluid rheological properties. As the accurate calculation (or measurement) of drilling fluid has been a major issue in the field, the topic of this work has both educational and informative aspects. I like how authors investigate and organize their works. I have one main question and several spell-checks as follows.

Q) It seems that "real-time" does not mean "in-situ". Indeed, it has been required tremendously to measure the in-situ drilling fluid property "during drilling". Among the four methods you listed, only "pipe viscometer" may have possibilities for it. However, as far as I understand, only AV(Apparent Viscosity) at the pipe wall could be back-calculated from the measured dp/dL. 

1) How do we know average viscosity at a certain point of pipe not only near the pipe wall? It can be apparent viscosity but the average value is important to predict CTR(cutting transport ratio). We know rpm of drill pipe so may be able to obtain velocity profile from N-S equation. Then, from shear rate at the wall, we may have shear stress profile. Is there any paper you found about this?

2) Changing question 1 a little bit, how do we obtain shear stress vs. shear rate plot from pipe viscometer? This question is similar to the above one.

[Typo]

  1. pp4. line 132; sonsor -> sensor
  2. pp.5 line 195; pip -> pipe
  3. pp.5 line 196; pip -> pipe

Sincerely,

Round 2

Reviewer 1 Report

Dear Authors, thanks for addressing my comments in detail. I do not have further comments.